# Ag Nanoparticles Decorated CuO@RF Core-Shell Nanowires for High-Performance Surface-Enhanced Raman Spectroscopy Application

**DOI:** 10.3390/molecules27238460

**Published:** 2022-12-02

**Authors:** Tung-Hao Chang, Hsin-Wei Di, Yu-Cheng Chang, Chia-Man Chou

**Affiliations:** 1Department of Radiation Oncology, Changhua Christian Hospital, Changhua 50006, Taiwan; 2Department of Radiological Technology, Yuanpei University, Hsinchu 30015, Taiwan; 3Department of Medical Imaging and Radiological Sciences, Central Taiwan University of Science and Technology, Taichung 40601, Taiwan; 4Department of Materials Science and Engineering, Feng Chia University, Taichung 407102, Taiwan; 5Department of Surgery, Taichung Veterans General Hospital, Taichung 40705, Taiwan; 6College of Medicine, National Yang Ming Chiao Tung University, Taipei 11221, Taiwan; 7Department of Post-Baccalaureate Medicine, National Chung Hsing University, Taichung 40227, Taiwan

**Keywords:** CuO nanowires, thermal oxidation process, resorcinol–formaldehyde, CuO@RF@Ag core-shell nanowires, rhodamine 6G, amoxicillin, 5-fluorouracil

## Abstract

Vertical-aligned CuO nanowires have been directly fabricated on Cu foil through a facile thermal oxidation process by a hotplate at 550 °C for 6 h under ambient conditions. The intermediate layer of resorcinol–formaldehyde (RF) and silver (Ag) nanoparticles can be sequentially deposited on Cu nanowires to form CuO@RF@Ag core-shell nanowires by a two-step wet chemical approach. The appropriate resorcinol weight and silver nitrate concentration can be favorable to grow the CuO@RF@Ag nanowires with higher surface-enhanced Raman scattering (SERS) enhancement for detecting rhodamine 6G (R6G) molecules. Compared with CuO@Ag nanowires grown by ion sputtering, CuO@RF@Ag nanowires exhibited a higher SERS enhancement factor of 5.33 × 10^8^ and a lower detection limit (10^−12^ M) for detecting R6G molecules. This result is ascribed to the CuO@RF@Ag nanowires with higher-density hot spots and surface-active sites for enhanced high SERS enhancement, good reproducibility, and uniformity. Furthermore, the CuO@RF@Ag nanowires can also reveal a high-sensitivity SERS-active substrate for detecting amoxicillin (10^−10^ M) and 5-fluorouracil (10^−7^ M). CuO@RF@Ag nanowires exhibit a simple fabrication process, high SERS sensitivity, high reproducibility, high uniformity, and low detection limit, which are helpful for the practical application of SERS in different fields.

## 1. Introduction

Presently, the detection equipment commonly used in the detection of drugs is quite expensive and time-consuming, so it is necessary to develop a simple, fast, reliable, and highly sensitive detection system [1,2]. Surface-enhanced Raman scattering (SERS) technology can detect low concentrations of chemicals, biomolecules, or heavy metals due to its high selectivity and sensitivity without sample pretreatment [3,4]. Furthermore, SERS is a versatile analytical technique that can further increase the intensity of the Raman-enhancing effect by adsorbing the number of molecules on the surface of metal nanostructures [5,6,7]. So far, the SERS technique has been recognized as one of the most sensitive and promising analytical tools in the chemical, biological, environmental, and clinical fields because it can provide much structural information [8,9].

In most cases, two main mechanisms have been adopted to explain the enhancement effect of SERS. One is the electromagnetic mechanism, in which excitation occurs in localized surface plasmon resonances on noble metal surfaces, and the electric field is amplified by an enhancement factor of 10^6^ [3,10,11]. The other is that the chemical mechanism, which belongs to the charge transfer between the molecule and the substrate surface, significantly amplifies the molecular polarization tensor and improves the Raman enhancement [12,13,14]. SERS substrates with noble metal nanostructures can be efficiently tuned for the electromagnetic mechanism by controlling their size, shape, and distribution on suitable substrates [9,15,16,17]. However, several problems limit the development of SERS substrates, such as low nanostructure density, uncontrolled aggregation, and uneven distribution of nanostructures [18,19].

Recently, one-dimensional semiconductor nanostructures with decorated noble metal nanoparticles to expand the arrangement of hot spots along three dimensions have provided suitable SERS substrates due to their excellent detection sensitivity, stability, uniformity, and reusability [20,21]. Previous studies have used thermal evaporation or ion-sputtering methods to uniformly deposit noble metal nanoparticles on one-dimensional semiconductor nanostructures for SERS applications [18,19,22,23,24]. Although these methods can be used to prepare high-performance SERS substrates, they do not contribute to energy saving and carbon reduction because the process needs to be carried out under high vacuum conditions. In addition, less literature reported the fabrication of SERS substrate on one-dimensional semiconductor nanostructures by coating a layer of resorcinol–formaldehyde (RF) resin to reduce noble metal nanoparticles [25]. In previous studies, resorcinol may serve multiple functions: it can act as a reactant to form an RF layer and passivate the surface of metal nanoparticles to prevent them from agglomerating [26]. Furthermore, resorcinol can also act as a reducing agent to reduce metal salts to metal nanoparticles [27].

In this study, we present a facile method for fabricating three-dimensional (3D) CuO@RF@Ag core-shell nanowires on Cu foil, which can detect multiple chemicals with a facile and cost-effective method. This 3D SERS substrate exhibited ultra-sensitivity for detecting various types of molecules, e.g., R6G, amoxicillin, and 5-fluorouracil, simultaneously suggesting its generality. Furthermore, the low detection limit of CuO@RF@Ag nanowires can be attained for R6G (10^−12^ M), amoxicillin (10^−10^ M), and 5-fluorouracil (10^−7^ M). This work demonstrates the promising use of CuO@RF@Ag nanowires with high SERS sensitivity, high reproducibility, high uniformity, and low detection limit, which shall be beneficial to the practical application of SERS in different fields.

## 2. Results and Discussion

Figure 1 shows a schematic illustration of the fabrication process used to grow CuO@RF@Ag nanowires. First, a thermal oxidation process was used to fabricate CuO nanowires on the Cu foil by a hotplate at 550 °C for 6 h under ambient conditions. Second, a self-assembly APTMS monolayer can raise the hydrophilic properties of CuO nanowires to decorate the uniform RF layer. Third, CuO nanowires with an RF layer can reduce the Ag^+^ to Ag for forming CuO@RF@Ag nanowires. Figure 2a,b reveal the tilt-view FESEM images of CuO nanowires grown with a hotplate and furnace at 550 °C for 6 h under ambient conditions. It can be observed that the hotplate has better uniformity than the furnace for the thermal oxidation of Cu foil to CuO nanowires. This result can be attributed to the closed furnace space and insufficient oxygen content during the thermal oxidation process. As a result, CuO nanowires cannot grow efficiently and uniformly on the Cu foil. Figure 2c,d reveal the cross-sectional FESEM images of CuO nanowires grown with a hotplate and furnace. The lengths of CuO nanowires are 9–12 µm and 23–31 µm, respectively. Although the CuO nanowires grown by thermal oxidation in a furnace have longer lengths than a hotplate, their uniformity and tip collapse may limit subsequent SERS applications. Therefore, the hotplate was chosen as the instrument for the thermal oxidation of Cu foil to form CuO nanowires.

The vertical-aligned CuO nanowires exhibit a suitable structure, which can benefit the uniform deposition of Ag nanoparticles to form CuO@Ag nanowires by an ion-beam sputtering system at a ~3.5 × 10^−6^ Torr pressure. In order to more effectively observe the microstructure, crystal structure, and composition of CuO@Ag nanowires with fabricated Ag nanoparticles on the CuO nanowires by an ion-sputtering method for 30 s, field-emission transmission electron microscopy (FETEM) analysis was further carried out in this study. The FETEM image in Figure 3a reveals that the CuO nanowire has been completely decorated with small Ag nanoparticles (2.5–12 nm). Furthermore, the high-angle annular dark field (HAADF) image (Figure 3b) can more clearly present the distribution of Ag nanoparticles on the CuO nanowire. Figure 3c shows the high-resolution FETEM image of a part of a CuO@Ag nanowire. There are two different lattice fringes with interplanar distances of 0.253 nm and 0.236 nm indexed to the (002) plane of monoclinic CuO (JCPDS Card No. 80-1917) and (111) plane of cubic Ag (JCPDS Card No. 89-3722), indicating that the crystal structure of CuO@Ag nanowire was composed by CuO and Ag. The corresponding energy-dispersive spectroscopy (EDS) mapping images (Figure 3d) of CuO@Ag nanowire are composed of Cu, O, and Ag, respectively. The XRD pattern of CuO@Ag nanowires is shown in Figure 3e. It can be seen that all diffraction peaks are attributed to cubic Cu crystals (JCPDS Card No. 85-1326), cubic Cu_2_O crystals (JCPDS Card No. 78-2076), and monoclinic CuO crystals (JCPDS Card No. 80-1917), respectively. Cu diffraction peaks come from Cu foil. The Cu_2_O diffraction peaks are ascribed to the interlayer between Cu foil and CuO nanowires. Typically, the thermal oxidation process produces a two-layer microstructure consisting of a Cu-rich Cu_2_O phase in contact with Cu and a top CuO layer [28]. The formation of CuO nanowires results from competition between the grain boundaries and lattice diffusion of Cu atoms on Cu_2_O [29].

Since the poor hydrophilicity of CuO nanowires makes the RF layer challenging to coat, even non-uniformly, self-assembled APTMS is used to increase its surface hydrophilicity. As a result, the water contact angle was significantly reduced from 23.43° to 8.1° after the self-assembly APTMS layer on CuO nanowires, as shown in Appendix A. Figure 4a reveals the tilt-view FESEM images of CuO@RF nanowires grown with different weights of resorcinol at 90 °C for 30 min under vigorous stirring. The different weights of resorcinol are 32.74 (no APTMS), 16.37, 32.74, and 65.48 mg, respectively. Compared with the no-self-assembly APTMS layer on the CuO nanowires, the RF layer cannot be uniformly deposited on the CuO nanowires and easily form a film on the top. This result also proves that the hydrophilicity of CuO nanowires can be significantly increased by the self-assembly APTMS layer, which is beneficial to the uniform coating of the RF layer. In addition, the surface of the CuO nanowires presents more particles and is more inhomogeneous with the increase in resorcinol weight. Therefore, the weight of resorcinol is selected to be 16.37 mg in subsequent experiments. Figure 4b reveals the tilt-view FESEM images of CuO@RF nanowires immersed in the different concentrations of AgNO_3_ solution for 2 h under vigorous stirring. The concentrations of the AgNO_3_ solution are 0.15, 0.30, 0.45, and 0.60 mM, respectively. As the concentration gradually increased, the size of the Ag nanoparticles also gradually increased. When the concentration reached 0.60 mM, Ag nanoparticles even formed a thin film. In our previous work, the appropriate distribution of Ag nanoparticles could provide the best hot-spot distribution, effectively enhancing the electromagnetic field strength and producing stronger SERS enhancement [18].

Figure 5a,b show the FETEM and HAADF images of a CuO@RF@Ag nanowire fabricated at the resorcinol weight of 16.37 mg and AgNO_3_ concentration of 0.45 mM. These results show that a CuO nanowire has been completely decorated with high-density Ag nanoparticles. The high-resolution FETEM image (Figure 5c) is a part of a CuO@RF@Ag nanowire. An RF layer with a thickness of about 2.5 nm can be observed on the surface of the CuO nanowire. In addition, a lattice fringe with an interplanar distance of 0.236 nm is indexed to the (111) plane of cubic Ag (JCPDS Card No. 89-3722). In Appendix A, the XRD diffraction pattern of CuO@RF@Ag nanowires was grown at a resorcinol weight of 16.37 mg and a AgNO_3_ concentration of 0.45 mM. The XRD diffraction peaks of Ag cannot be observed in Appendix A. This phenomenon shall be ascribed to the sizes of Ag nanoparticles being too small to detect XRD peaks. The EDS mapping images (Figure 5d) can also clearly depict the arrangement of Ag nanoparticles, while Cu, O, and Ag are homogeneously distributed on a CuO@RF@Ag nanowire. The EDS spectrum (Figure 5e) of a CuO@RF@Ag nanowire with Cu, O, Ag, and Au (Au is from the TEM grid) can be used to evaluate the elemental analysis. This result proves that there are no other impurity elements in the CuO@RF@Ag nanowire.

Figure 6a reveals the SERS spectra of CuO nanowires deposited Ag nanoparticles at different sputtering times and immersed in the R6G solution (10^−6^ M) for 1 h. The sputtering times are 15, 30, 60, and 90 s, respectively. The primary vibrational mode for the characteristic peaks of the R6G molecule is C-H in-plane bending mode (1127 and 1186 cm^−1^), C−O−C in-plane bending (1311 cm^−1^), and aromatic C−C stretch (1362, 1421, 1510, 1572, and 1649 cm^−1^), respectively [30]. It can be ascertained that the strongest Raman signal is at the sputtering time of 30 s. As the sputtering time increases, the Raman signal tends to decrease gradually. This possible reason shall be ascribed to providing a better distribution of hot spots at a sputtering time of 30 s, thereby significantly improving the intensity of its Raman signal. Figure 6b reveals the SERS spectra of CuO@RF nanowires-deposited Ag nanoparticles at different concentrations of the AgNO_3_ solution and immersed in the R6G solution (10^−6^ M) for 1 h. The concentrations of the AgNO_3_ solution are 0.15, 0.30, 0.45, and 0.60 mM, respectively. The Raman signal tends to increase gradually with the increase of AgNO_3_ concentration, and then the strongest Raman signal is at the concentration of 0.45 mM. If the concentration continues to increase, the Raman signal tends to decrease gradually. When the concentration is 0.45 mM, it can provide a better distribution of hot spots, thus significantly improving the intensity of its Raman signal. Appendix A shows the Raman spectrum of CuO@RF nanowires. Due to the relatively weak Raman signal, it should not affect the measurement of other chemicals. This result is consistent with FESEM image observation (Figure 4b), and the subsequent measurement is discussed under this condition. According to previous reports, the SERS enhancement factor (EF) values of 1649 cm^−1^ for CuO@Ag nanowires (30 s) and CuO@RF@Ag nanowires (0.45 mM) were calculated to be 3.09 × 10^8^ and 5.33 × 10^8^, respectively [31,32,33]. The SERS EF of CuO@RF@Ag nanowires is 1.72 times higher than the CuO@Ag nanowires.

The uniformity and reproducibility of SERS substrates play significant roles in practical SERS applications [34,35]. For example, Figure 7a,b show the SERS spectra of R6G (10^−6^ M) at ten random points on CuO@Ag nanowires (30 s) and CuO@RF@Ag nanowires (0.45 mM), respectively. The slight relative standard deviations (RSDs) of 6.94% (CuO@Ag nanowires) and 7.47% (CuO@RF@Ag nanowires) were obtained for the notable peaks at 1649 cm^−1^. The small RSD indicates that CuO@Ag nanowires or CuO@RF@Ag nanowires can be used as reliable SERS substrates with high uniformity. In order to investigate the reusability of CuO@Ag nanowires (30 s) and CuO@RF@Ag nanowires (0.45 mM), three cycles of immersion–detection and removing processes were used to evaluate the SERS signal of R6G (10^−7^ M), as shown in Figure 7c,d. The removal process irradiates UV light (253.7 nm, 36 W) for 1 h to photodegrade R6G molecules. These results show that the CuO@Ag nanowires (30 s) and CuO@RF@Ag nanowires (0.45 mM) exhibit efficient reusability and still reveal similar Raman signal intensities after three cycles.

Figure 8a,b show the SERS spectra of different concentrations of the R6G solution on CuO@Ag nanowires (30 s) and CuO@RF@Ag nanowires (0.45 mM), respectively. For CuO@Ag nanowires, the intensity of the characteristic peaks of R6G increased significantly with the increase in the concentration, and the minimum detection concentration of R6G reached 10^−8^ M. For CuO@RF@Ag nanowires, the intensity of the characteristic peaks of R6G also increased significantly with the increase in the concentration, and the minimum detection concentration of R6G reached 10^−12^ M. Appendix A shows the Raman spectrum of R6G (10^−13^ M) for CuO@RF@Ag nanowires. No signal exhibits three times the signal-to-noise at an R6G concentration of 10^−13^ M. The SERS EF of CuO@RF@Ag nanowires is only 1.72 times higher than that of CuO@Ag nanowires. However, the lowest detection limit of CuO@RF@Ag nanowires can be significantly improved for four orders compared with CuO@Ag nanowires. Appendix A shows the concentration dependence of the R6G peak intensity at 1649 cm^−1^ as a function of R6G concentrations ranging from 10^−9^ to 10^−12^ M (R^2^ = 0.99886). This result also confirms that the CuO@RF@Ag nanowires have high sensitivity to the detection of low-concentration chemicals.

Herein, we selected two commercial drugs (amoxicillin and 5-fluorouracil) to evaluate the suitability of CuO@RF@Ag nanowires for drug detection. Amoxicillin is a bactericidal β-lactam antibiotic drug molecule from the aminopenicillin family, which can be used to treat various diseases caused by bacterial infections and prevent bacterial growth [36,37]. 5-Fluorouracil (5-FU) is an antimetabolite antitumor drug used to treat various solid tumors, such as colorectal, rectal, breast, gastric, pancreatic, liver, and bladder cancer [19,38]. Figure 9a shows the SERS spectra of different concentrations of amoxicillin solution on CuO@RF@Ag nanowires (0.45 mM). The concentrations of the amoxicillin solution are from 10^−6^ to 10^−10^ M. The primary vibrational mode for the characteristic peaks of the amoxicillin molecule is plane deformation (790 cm^−1^), benzene ring breathing (865 cm^−1^), amine bending (935 cm^−1^), C-H stretching (1038 cm^−1^), plane deformation of benzene (1288 cm^−1^), twisting of amine (1351 cm^−1^), asymmetric bending of CH_3_ (1490 cm^−1^), C-C stretching (1603 cm^−1^), and N-H bending (1651 cm^−1^), respectively. The intensity of the characteristic peaks of amoxicillin increased significantly with the increase in the concentration, and the minimum detection concentration of amoxicillin reached 10^−10^ M. Figure 9b shows the SERS spectra of different concentrations of 5-fluorouracil solution on CuO@RF@Ag nanowires (0.45 mM). The concentrations of 5-fluorouracil solution are from 10^−3^ to 10^−7^ M. The primary vibrational mode for the characteristic peaks of the 5-fluorouracil molecule is pyrimidine ring breathing (786 cm^−1^), ring plus C-F stretching (1234 cm^−1^), ring plus C-H wagging (1335 cm^−1^), trigonal ring (1400 cm^−1^), and symmetric C=O stretching (1667 cm^−1^), respectively [15,39]. The intensity of the characteristic peaks of 5-fluorouracil also increased significantly with the increase in the concentration, and the minimum detection concentration of 5-fluorouracil reached 10^−7^ M. Different studies and detection limits for the determination of amoxicillin and 5-FU using different SERS substrates are listed in Appendix A [40,41,42,43,44,45]. This result shows that the CuO@RF@Ag nanowires are also beneficial for detecting low-concentration antibiotics and antitumor drugs. CuO@RF@Ag nanowires can exhibit an excellent SERS effect, which also benefits the subsequent application of different drugs in Raman detection.

## 3. Materials and Methods

### 3.1. Materials

Copper foil (0.025mm thick, 99.8%) was commercially obtained from Alfa Aesar (Ward Hill, MA, USA). All chemicals were purchased from commercial sources and used without further purification. (3-Aminopropyl)triethoxysilane (APTMS, 95%, Acros, Geel, Belgium), resorcinol (C_6_H_6_O_2_, 99%, Alfa Aesar, USA), ammonium hydroxide (NH_4_OH, 28%, Alfa Aesar, USA), formaldehyde solution (CH_2_O, 37%, Merck, Darmstadt, Germany), silver nitrate (AgNO_3_, 99%, Alfa Aesar, USA), hydrochloric acid (HCl, 37%, Sigma-Aldrich, Darmstadt, Germany), rhodamine 6G (R6G, 99.8%, Acros, Germany), amoxicillin (50 mg/mL, Y F Chemical, New Taipei City, Taiwan), 5-fluorouracil (5FU, 50 mg/mL, Nang Kuang Pharmaceutical, Tainan, Taiwan), and ethanol (C_2_H_5_OH, 99%, Sigma-Aldrich, Darmstadt, Germany) were used in this experiments. De-ionized water with a resistivity higher than 18.2 MΩ was used for all solution preparations.

### 3.2. Fabrication of CuO Nanowires

A copper foil was cut into 1 cm^2^ square substrates. First, the Cu substrates were cleaned with dilute HCl to remove the surface oxide layer and adsorbed impurities. Then, the Cu substrates were cleaned with acetone, de-ionized water, and ethanol in an ultrasonic bath for 10 min and dried in the air. The thermal oxidation method can be used to directly grow CuO nanowires on the Cu foil with a facile hotplate (Corning, PC-420D, Corning, New York, NY, USA). The reaction temperatures are 550 °C for 6 h under ambient conditions.

### 3.3. Fabrication of CuO@Ag Core-Shell Nanowires

The substrates with GuO nanowires deposited Ag nanoparticles to form CuO@Ag core-shell nanowires by an ion-beam sputtering system at a ~3.5 × 10^−6^ Torr pressure for the different deposition times.

### 3.4. Fabrication of CuO@RF@Ag Core-Shell Nanowires

The substrates with GuO nanowires were immersed in an ethanol solution of 50 mL containing 5 mM APTMS for 12 h at room temperature to increase the hydrophilicity of the substrate surface. Next, the substrates were rinsed with ethanol and de-ionized water several times and dried under an air purge. For the growth of the RF layer on the CuO nanowires, the substrates were immersed in a 40 mL aqueous solution containing the different weights of resorcinol, 0.0284 mL ammonium hydroxide, and 0.0444 mL formaldehyde, which was vigorous stirred for 30 min at 90 °C. Next, the substrate was rinsed with ethanol several times and dried at 60 °C for 2 h. For the deposition of Ag nanoparticles on the CuO@RF core-shell nanowires, the substrate was immersed in an aqueous solution of 50 mL containing the different concentrations of AgNO_3_ under vigorous stirring for 2 h. Finally, the product substrate was rinsed with ethanol several times and dried at 60 °C for 2 h.

### 3.5. Material Characterization

Field-emission scanning electron microscope (FESEM) images were obtained on a Hitachi S-4800 (Hitachi, Tokyo, Japan) operated at 15 kV. In addition, field-emission transmission electron microscopy (FETEM) images, energy dispersive spectrometry (EDS) mapping images and spectrum were obtained on a JEOL JEM-2100F (JEOL, Tokyo, Japan) operated at 200 kV. Furthermore, the X-ray powder diffraction (XRD) spectrum was measured on a Bruker D2 phaser (Bruker, Billerica, MA, USA) X-ray diffractometer equipped with Cu Kα radiation (λ = 0.154060 nm), employing a scanning rate of 0.04° s^−1^.

### 3.6. SERS Measurements

A confocal Raman microscope (MRI532S, Protrustech, Tainan, Taiwan) was used to measure the Raman spectra operated at an excitation wavelength of 532 nm. For the Raman measurement conditions, the laser irradiation power and the detector integration time were 1 mW and 0.15 s, respectively. In addition, the rhodamine 6G (R6G) molecule was used as a chemical target to evaluate the SERS properties of as-prepared substrates. In addition, the amoxicillin and 5-fluorouracil molecules were used as the drug targets to evaluate the SERS properties of CuO@RF@Ag core-shell nanowires. The SERS substrates were immersed in target solutions of different concentrations for 1 h at room temperature in the dark and then dried by air blowing.

## 4. Conclusions

In this study, we report a facile thermal oxidation process to fabricate vertically aligned CuO nanowires on Cu foil with a hotplate at 550 °C for 6 h under ambient conditions. In addition, the self-assembly of the APTMS layer can improve the hydrophilic properties of CuO nanowires and contribute to the uniform coating of the RF layer. The appropriate resorcinol weight and AgNO_3_ concentration can be beneficial to growing the CuO@RF@Ag nanowires with higher SERS enhancement for detecting R6G molecules. Furthermore, different positions and reusability experiments prove that the CuO@RF@Ag nanowires exhibit excellent uniformity and stability. The CuO@RF@Ag nanowires can also exhibit a superior SERS-active substrate for detecting the lower concentration of R6G (10^−12^ M), amoxicillin (10^−10^ M), and 5-fluorouracil (10^−7^ M). These results promise to apply SERS technology to rapid low-concentration detection in different chemical or drug fields.

## Figures and Tables

**Figure 1 molecules-27-08460-f001:**
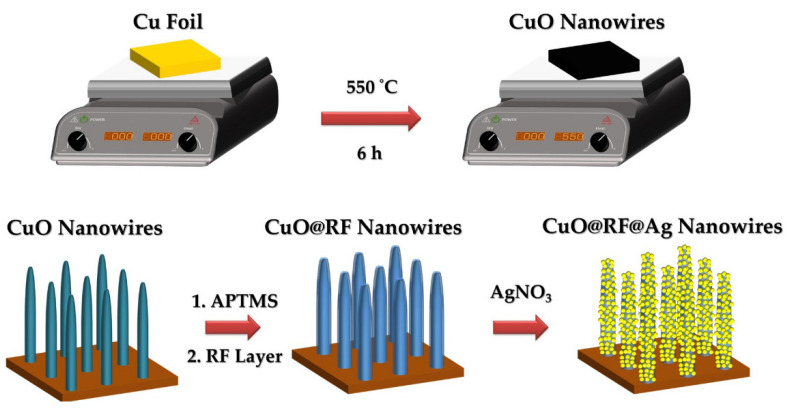
Schematic diagram of the preparation process for CuO@RF@Ag nanowires.

**Figure 2 molecules-27-08460-f002:**
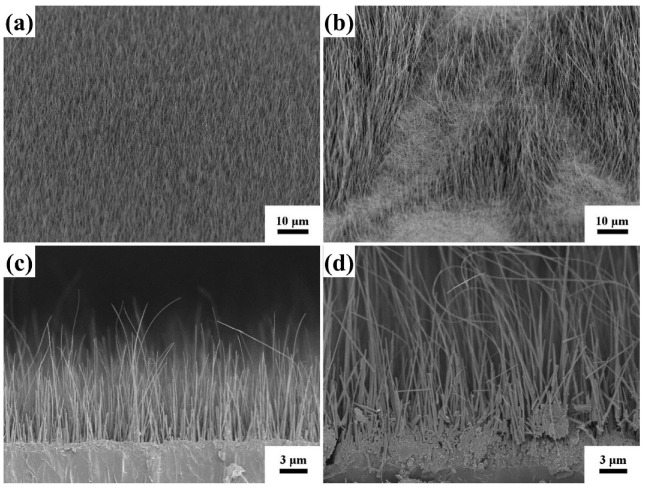
The tilt-view FESEM images of CuO nanowires grown on Cu foil with a (**a**) hotplate and (**b**) furnace at 550 °C for 6 h under ambient conditions. The cross-sectional FESEM images of CuO nanowires grown on Cu foil with a (**c**) hotplate and (**d**) furnace at 550 °C for 6 h under ambient conditions.

**Figure 3 molecules-27-08460-f003:**
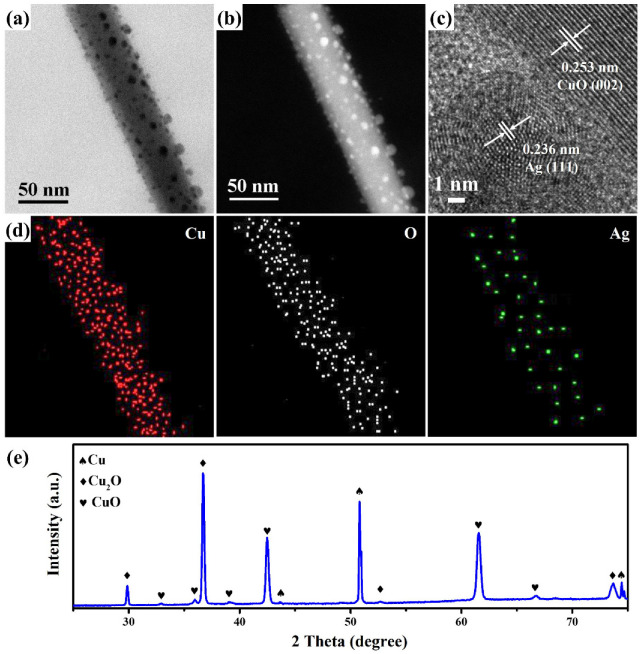
The (**a**) FETEM, (**b**) HAADF, (**c**) HRTEM, and (**d**) EDS mapping images of a CuO@Ag nanowire grown by an ion-sputtering method for 30 s. (**e**) XRD diffraction pattern of CuO@Ag nanowires grown by an ion-sputtering method for 30 s.

**Figure 4 molecules-27-08460-f004:**
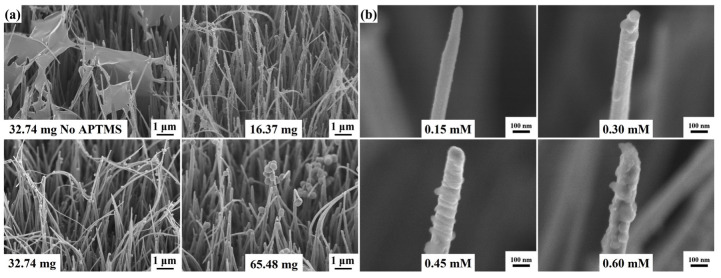
(**a**) The tilt-view FESEM images of CuO@RF nanowires grown at the different weights of resorcinol. The different weights of resorcinol are 32.74 (no APTMS), 16.37, 32.74, and 65.48 mg, respectively. (**b**) The tilt-view FESEM images of CuO@RF@Ag nanowires grown at different concentrations of AgNO_3_. The concentrations of AgNO_3_ are 0.15, 0.30, 0.45, and 0.60 mM, respectively.

**Figure 5 molecules-27-08460-f005:**
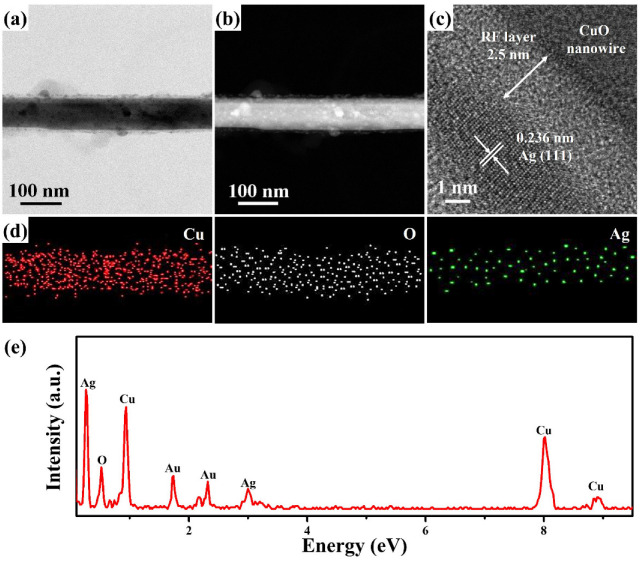
The (**a**) FETEM, (**b**) HAADF, (**c**) HRTEM, (**d**) EDS mapping images, and (**e**) EDS spectrum of a CuO@Ag nanowire fabricated at a resorcinol weight of 16.37 mg and a AgNO_3_ concentration of 0.45 mM.

**Figure 6 molecules-27-08460-f006:**
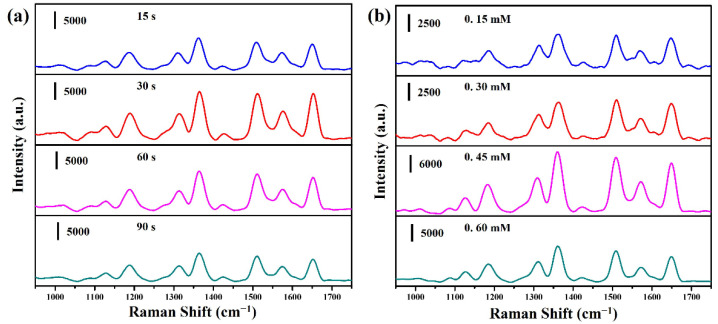
(**a**) SERS spectra of R6G (10^−6^ M) on CuO@Ag nanowires grown at different sputtering times. (**b**) SERS spectra of R6G (10^−6^ M) on CuO@RF@Ag nanowires grown at different concentrations of AgNO_3_ solution.

**Figure 7 molecules-27-08460-f007:**
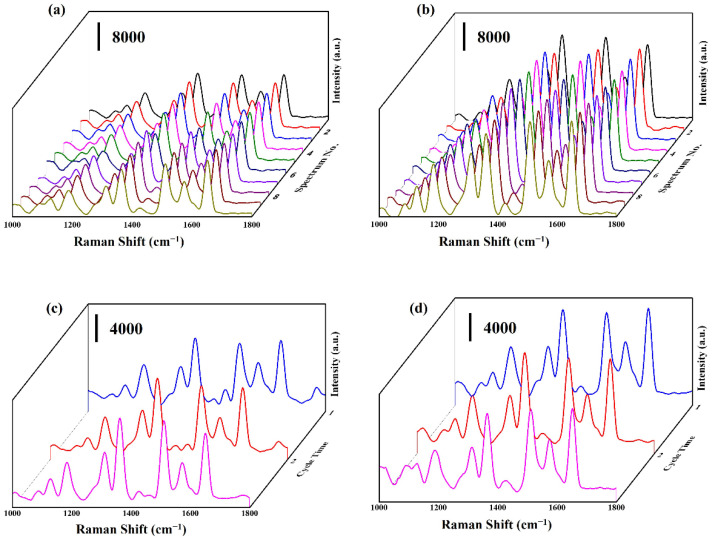
SERS spectra of R6G (10^−6^ M) at ten random points on (**a**) CuO@Ag nanowires and (**b**) CuO@RF@Ag nanowires. SERS spectra of R6G (10^−7^ M) on (**c**) CuO@Ag nanowires, and (**d**) CuO@RF@Ag nanowires at the three cycles for 1 cycle (blue line), 2 cycle (red line), and 3 cycle (purple).

**Figure 8 molecules-27-08460-f008:**
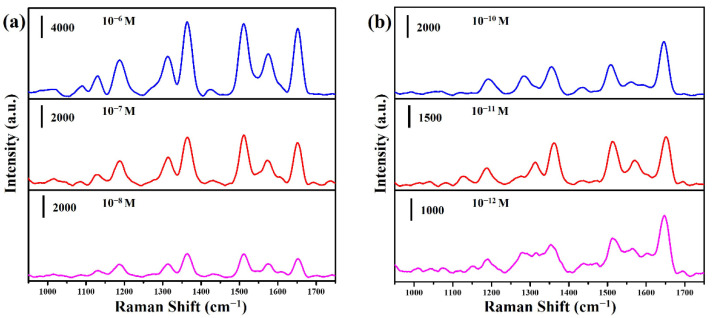
SERS spectra of R6G at different concentrations were obtained from (**a**) CuO@Ag nanowires and (**b**) CuO@RF@Ag nanowires.

**Figure 9 molecules-27-08460-f009:**
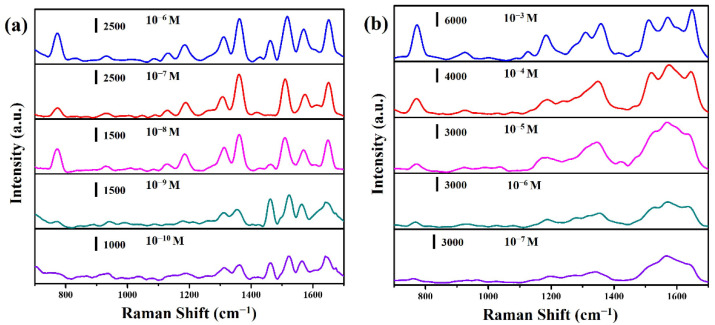
SERS spectra of (**a**) amoxicillin and (**b**) 5-fluorouracil at different concentrations were obtained from CuO@RF@Ag nanowires.

## Data Availability

No new data were created or analyzed in this study. Data sharing is not applicable to this article.

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
