# Peer review of "Ag Nanoparticles Decorated CuO@RF Core-Shell Nanowires for High-Performance Surface-Enhanced Raman Spectroscopy Application"

_molecules, 2022, doi:10.3390/molecules27238460_

Round 1
Reviewer 1 Report
In this manuscript, the authors demonstrate a synthesis of CuO@RF@Ag core-shell nanowires for SERS detection of molecules. Morphologies and crystal structures of CuO@Ag and CuO@RF@Ag nanowires were analyzed. The SERS spectra of R6G molecules on both nanowires were compared with respect to the molecular concentrations and Ag layer thicknesses. Data of other molecules - amoxicillin and 5-fluorouracil were also demonstrated.
The manuscript has serious flaws in presenting the detection range of molecules and data analysis. No quantitative comparison between conditions was shown, nor the control experiments for target molecules regarding the detection ranges were presented.
Overall, the results are not conclusive and original enough to be published in the journal Molecule.
Author Response
In this manuscript, the authors demonstrate a synthesis of CuO@RF@Ag core-shell nanowires for SERS detection of molecules. Morphologies and crystal structures of CuO@Ag and CuO@RF@Ag nanowires were analyzed. The SERS spectra of R6G molecules on both nanowires were compared with respect to the molecular concentrations and Ag layer thicknesses. Data of other molecules - amoxicillin and 5-fluorouracil were also demonstrated.
The manuscript has serious flaws in presenting the detection range of molecules and data analysis. No quantitative comparison between conditions was shown, nor the control experiments for target molecules regarding the detection ranges were presented.
We have provided the Raman spectrum of CuO@RF nanowires. Due to the relatively weak Raman signal, it should not affect the measurement of other chemicals.
We also have provided the concentration v.s. signal intensity (1649 cm−1) in Figure S4. Figure S4 shows the concentration dependence of R6G peak intensity at 1649 cm−1 as a function of R6G concentrations ranging from 10−9 to 10−12 M. (R2 = 0.99886 )
Overall, the results are not conclusive and original enough to be published in the journal Molecule.
There are eight significant findings in the present study:
- Vertical-aligned CuO nanowires have been directly fabricated on the Cu foil through a facile thermal oxidation process by a hotplate at 550 °C for 6 h under ambient conditions.
- The self-assembly of the APTMS layer can improve the hydrophilic properties of CuO nanowires and contribute to the uniform coating of the RF layer.
- The intermediate layer of resorcinol-formaldehyde (RF) and silver (Ag) nanoparticles can be sequentially deposited on Cu nanowires to form CuO@RF@Ag core-shell nanowires by a two-step wet chemical approach.
- The appropriate resorcinol weight and silver nitrate concentration can be favorable to grow the CuO@RF@Ag nanowires with higher surface-enhanced Raman scattering (SERS) enhancement for detecting rhodamine 6G (R6G) molecules.
- Compared with CuO@Ag nanowires grown by ion sputtering, CuO@RF@Ag nanowires exhibited a higher SERS enhancement factor of 5.33 × 108 and a lower detection limit (10−12 M) for detecting R6G molecules.
- This result is ascribed to the CuO@RF@Ag nanowires with higher-density hot spots and surface-active sites for enhanced high SERS enhancement, good reproducibility, and uniformity.
- Furthermore, the CuO@RF@Ag nanowires can also reveal a high-sensitivity SERS-active substrate for detecting amoxicillin (10−10 M) and 5-fluorouracil (10−7 M).
- CuO@RF@Ag nanowires exhibit a simple fabrication process, high SERS sensitivity, high reproducibility, high uniformity, and low detection limit, which are helpful for the practical application of SERS in different fields.

Reviewer 2 Report
The paper reports on the preparation, characterization and SERS application of CuO nanowires decorated with Ag nanoparticles.
Given that this kind of nanostructures has already been described in literature (https://doi.org/10.1021/acsomega.0c02301, DOI:10.1007/s11468-020-01358-6, https://doi.org/10.1002/admi.202200047), it is particularly important that authors well clarify the specific novelty of the presented results. In fact, the main flaw of the manuscript is the lack of clarity on its subject.
Actually, both the title and the abstract are focused on the use of an intermediate layer of resorcinol-formaldehyde to deposit Ag nanoparticles as an efficient and innovative chemical way to prepare new nanowires, indicated as CuO@RF@Ag NWs, and characterized by a higher enhancement factor with respect to other similar structures. However, half of the paper is devoted to the characterization and application of simpler and more standard CuO@Ag NWs. This inconsistency between title and text creates confusion about the true purpose of the paper. Differently, if the goal of the paper is to compare the two structures, the title must be changed accordingly, and information about the preparation of the CuO@Ag NWs, briefly cited in the text as obtained by “ion sputtering”, must be added to the materials and methods section, which is presently just dedicated to CuO@RF@Ag NWs.
Some less relevant point must be also taken into consideration during revision:
- Refs 18 and 19 are cited as the source of information about the SERS properties of CuO@Ag NWs, but the two papers report on ZnO structures;
- There are some typos to be corrected, such as cut sentences (“To verify that CuO@RF@Ag nanowires can still exhibit higher sensitivity for detect-232 ing low concentrations of different drugs.”), or wrongly organized sentences (“Figure 3c shows that the high-resolution 110 FETEM image is a part of a CuO@Ag nanowire.” Which must be probably written as “Figure 3c shows the high-resolution 110 FETEM image of a part of a CuO@Ag nanowire.”)
- It would be important to better explain how the minimum detection concentration of R6G, indisted as 10-8 M) has been determined.
Author Response
The paper reports on the preparation, characterization and SERS application of CuO nanowires decorated with Ag nanoparticles.
Given that this kind of nanostructures has already been described in literature (https://doi.org/10.1021/acsomega.0c02301, DOI:10.1007/s11468-020-01358-6, https://doi.org/10.1002/admi.202200047), it is particularly important that authors well clarify the specific novelty of the presented results. In fact, the main flaw of the manuscript is the lack of clarity on its subject.
Response: Thanks for your reminder. These references used ion-sputtering or electron beam evaporation to uniformly deposit noble metal nanoparticles on CuO nanostructures for SERS applications. Although these methods can be used to prepare high-performance SERS substrates, they do not contribute to energy saving and carbon reduction because the process needs to be carried out under high vacuum conditions. In this study, we present a facile method for fabricating three-dimensional (3D) CuO@RF@Ag core-shell nanowires on the Cu foil, which can detect multiple chemicals by a facile and cost-effective method. Furthermore, this 3D SERS substrate exhibited an ultra-sensitivity for detecting various types of molecules, e.g., R6G, amoxicillin, and 5-fluorouracil, simultaneously suggesting its generality. Furthermore, the low detection limit of CuO@RF@Ag nanowires can be attained for R6G (10−12 M), amoxicillin (10−10 M), and 5-fluorouracil (10−7 M).
Actually, both the title and the abstract are focused on the use of an intermediate layer of resorcinol-formaldehyde to deposit Ag nanoparticles as an efficient and innovative chemical way to prepare new nanowires, indicated as CuO@RF@Ag NWs, and characterized by a higher enhancement factor with respect to other similar structures. However, half of the paper is devoted to the characterization and application of simpler and more standard CuO@Ag NWs. This inconsistency between title and text creates confusion about the true purpose of the paper. Differently, if the goal of the paper is to compare the two structures, the title must be changed accordingly, and information about the preparation of the CuO@Ag NWs, briefly cited in the text as obtained by “ion sputtering”, must be added to the materials and methods section, which is presently just dedicated to CuO@RF@Ag NWs.
Response: Thanks for your reminder. We have added these descriptions of CuO@Ag nanowires in the revised manuscript.
Some less relevant point must be also taken into consideration during revision:
- Refs 18 and 19 are cited as the source of information about the SERS properties of CuO@Ag NWs, but the two papers report on ZnO structures;
Response: Thanks for your reminder. We have added three references (Ref. 31-33) about CuO@Ag in the revised manuscript.
- There are some typos to be corrected, such as cut sentences (“To verify that CuO@RF@Ag nanowires can still exhibit higher sensitivity for detect-232 ing low concentrations of different drugs.”), or wrongly organized sentences (“Figure 3c shows that the high-resolution 110 FETEM image is a part of a CuO@Ag nanowire.” Which must be probably written as “Figure 3c shows the high-resolution 110 FETEM image of a part of a CuO@Ag nanowire.”)
Response: Thanks for your reminder. These sentences have been amended in the revised manuscript.
- It would be important to better explain how the minimum detection concentration of R6G, indisted as 10-8 M) has been determined.
Response: Thanks for your reminder. The detection limit is determined by 3 times signal to noise in the SERS spectra. No signal is 3 times signal to noise at the R6G concentration of 10−9 M.

Reviewer 3 Report
The Manuscript “Ag Nanoparticles Decorated CuO@RF Core-Shell Nanowires for High-Performance Surface-Enhanced Raman Spectroscopy Application” report a wet-chemistry method for obtaining high-performance SERS substrate based on CuO nanowires. Although preparation of such a composite is rather tedious, the obtained material showed superior analytical properties (sensitivity, reproducibility and stability under recycling) compared to the simple sputtering of silver over nanowires. The prepared composite is well characterized, research design is clear and results are quite promising. However, I have some questions and remarks, so the verdict is “Minor revision”.
First of all, please provide reasoning for such a complex four-component CuO-nanowires@APTMS@RF@AgNPs composite design in Introduction section, as it seems rather overcomplicated.
It’s incorrect to call powder XRD plot “spectrum” (Fig.3 label) as it depicts signal intensity vs. diffraction angle, which is not energy. It’s is better to call them “diffraction patterns”. Also, why there are no signals of silver in sputtered Ag@Cu composites XRD (Fig. 3,e)? Please, reconsider the attribution of signals at ~38° and ~44° 2θ as they also may belong to (111) and (200) reflexes of Fd3m silver. Finally, provide XRD pattern for CuO@RF@Ag composite as wet-chemical synthesis of AgNPs can potentially led to Ag2O formation as well.
Report contact angle measurement procedure for Cu nanowires with and w/o APTMS.
APTMS acronym usage is inconsistent (e.g. "aptems" in s.138); Please, bring it to uniformity.
Can organic components (APTMS and RF) be seen in Raman spectra of the composites? If yes, do they overlap with the important vibration band of commonly used analyte?
I have some notes on SERS data presentation:
-
Intensity vs Ag concentration plots (Fig.6) are hard to compare visually, please change plot type to waterfall or show them all at one panel.
-
Please, provide analytes’ concentration vs. signal intensity plots (for most convenient Raman peak) to show the linearity range and error estimation.
Author Response
Reviewers' comments:
The Manuscript “Ag Nanoparticles Decorated CuO@RF Core-Shell Nanowires for High-Performance Surface-Enhanced Raman Spectroscopy Application” report a wet-chemistry method for obtaining high-performance SERS substrate based on CuO nanowires. Although preparation of such a composite is rather tedious, the obtained material showed superior analytical properties (sensitivity, reproducibility and stability under recycling) compared to the simple sputtering of silver over nanowires. The prepared composite is well characterized, research design is clear and results are quite promising. However, I have some questions and remarks, so the verdict is “Minor revision”.
Response: Thanks for the pertinent and positive comments.
First of all, please provide reasoning for such a complex four-component CuO-nanowires@APTMS@RF@AgNPs composite design in Introduction section, as it seems rather overcomplicated.
Response: Thanks for your reminder. We have amended these descriptions in the revised manuscript. Previous studies have used thermal evaporation or ion-sputtering methods to uniformly deposit noble metal nanoparticles on one-dimensional semiconductor nanostructures for SERS applications. Although these methods can be used to prepare high-performance SERS substrates, they do not contribute to energy saving and carbon reduction because the process needs to be carried out under high vacuum conditions. In addition, less literature reported the fabrication of SERS substrate on one-dimensional semiconductor nanostructures by coating a layer of resorcinol-formaldehyde (RF) resin to reduce noble metal nanoparticles. In previous studies, resorcinol may serve multiple functions: it can act as a reactant to form RF layer and passivate the surface of metal nanoparticles to prevent them from agglomerating. Furthermore, resorcinol can also act as a reducing agent to reduce metal salts to metal nanoparticles.
It’s incorrect to call powder XRD plot “spectrum” (Fig.3 label) as it depicts signal intensity vs. diffraction angle, which is not energy. It’s is better to call them “diffraction patterns”. Also, why there are no signals of silver in sputtered Ag@Cu composites XRD (Fig. 3,e)? Please, reconsider the attribution of signals at ~38° and ~44° 2θ as they also may belong to (111) and (200) reflexes of Fd3m silver. Finally, provide XRD pattern for CuO@RF@Ag composite as wet-chemical synthesis of AgNPs can potentially led to Ag2O formation as well.
Response: Thanks for your reminder. We have amended this mistake in the revised manuscript. The diffraction peaks at 38.9° and 43.4° correspond to (200) and (111) planes of monoclinic CuO crystals (JCPDS Card No. 80–1917) and cubic Cu crystals (JCPDS Card No. 85–1326), respectively. We have indeed checked that there are no XRD peaks for Ag. This phenomenon shall be ascribed to the sizes of Ag nanoparticles being too small to detect XRD peaks. For example, figure S2 XRD diffraction pattern of CuO@RF@Ag nanowires was grown at the resorcinol weight of 16.37 mg and AgNO3 concentration of 0.45 mM. The XRD diffraction peaks of Ag or Ag2O cannot be observed in this Figure.
Report contact angle measurement procedure for Cu nanowires with and w/o APTMS.
Response: Thanks for your reminder. We have added the contact angle data in Figure S1.
APTMS acronym usage is inconsistent (e.g. "aptems" in s.138); Please, bring it to uniformity.
Response: Thanks for your reminder. We have amended this mistake in the revised manuscript.
Can organic components (APTMS and RF) be seen in Raman spectra of the composites? If yes, do they overlap with the important vibration band of commonly used analyte?
Response: Thanks for your reminder. Figure S3 shows the Raman spectrum of CuO@RF nanowires. Due to the relatively weak Raman signal, it should not affect the measurement of other chemicals.
I have some notes on SERS data presentation:
Intensity vs Ag concentration plots (Fig.6) are hard to compare visually, please change plot type to waterfall or show them all at one panel.
Response: Thanks for your reminder. We have changed the plot type to one panel in the revised manuscript.
Please, provide analytes’ concentration vs. signal intensity plots (for most convenient Raman peak) to show the linearity range and error estimation.
Response: Thanks for your reminder. We have provided the concentration v.s. signal intensity (1649 cm−1) in Figure S4. Figure S4 shows the concentration dependence of R6G peak intensity at 1649 cm−1 as a function of R6G concentrations ranging from 10−9 to 10−12 M. (R2 = 0.99886 )

Round 2
Reviewer 1 Report
The Raman spectrum of CuO@RF added by the authors still does not sufficiently support their conclusion. The authors must provide appropriate control experimental data (test results less than 10^-12 M) to confirm their lower detection limit.
If they conclude that CuO@RF@Ag NWs are beneficial for SERS, what is the point of showing data on CuO only?
Furthermore, the presentation of Figure S4 is completely wrong. There is no evidence that the detectable concentration range is down to 10^-12 M at all from the figure. Check the x-axis range.
It seems that the manuscript is still immature to be published in the journal Molecule.
Author Response
The Raman spectrum of CuO@RF added by the authors still does not sufficiently support their conclusion. The authors must provide appropriate control experimental data (test results less than 10^-12 M) to confirm their lower detection limit.
Response: Thanks for your reminder. We have added the Raman spectrum of R6G (10−13 M) for CuO@RF@Ag nanowires in Figure S5. No signal is 3 times signal to noise at the R6G concentration of 10−13 M.
If they conclude that CuO@RF@Ag NWs are beneficial for SERS, what is the point of showing data on CuO only?
Response: Thanks for your reminder. This research shows relevant data for CuO nanowires and in-depth analysis and discussion on preparing high-efficiency surface-enhanced Raman scattering substrates by depositing Ag nanoparticles through the ion-sputtering method. In addition, it can be verified that the CuO@RF@Ag nanowires can indeed exhibit better surface-enhanced Raman scattering substrates.
Furthermore, the presentation of Figure S4 is completely wrong. There is no evidence that the detectable concentration range is down to 10^-12 M at all from the figure. Check the x-axis range.
Response: Thanks for your reminder. We have labeled the detectable concentrations in Figure S4. However, due to the significant difference in concentration, it cannot be effectively presented.
It seems that the manuscript is still immature to be published in the journal Molecule.
There are eight significant findings in the present study:
- Vertical-aligned CuO nanowires have been directly fabricated on the Cu foil through a facile thermal oxidation process by a hotplate at 550 °C for 6 h under ambient conditions.
- The self-assembly of the APTMS layer can improve the hydrophilic properties of CuO nanowires and contribute to the uniform coating of the RF layer.
- The intermediate layer of resorcinol-formaldehyde (RF) and silver (Ag) nanoparticles can be sequentially deposited on Cu nanowires to form CuO@RF@Ag core-shell nanowires by a two-step wet chemical approach.
- The appropriate resorcinol weight and silver nitrate concentration can be favorable to grow the CuO@RF@Ag nanowires with higher surface-enhanced Raman scattering (SERS) enhancement for detecting rhodamine 6G (R6G) molecules.
- Compared with CuO@Ag nanowires grown by ion sputtering, CuO@RF@Ag nanowires exhibited a higher SERS enhancement factor of 5.33 × 108 and a lower detection limit (10−12 M) for detecting R6G molecules.
- This result is ascribed to the CuO@RF@Ag nanowires with higher-density hot spots and surface-active sites for enhanced high SERS enhancement, good reproducibility, and uniformity.
- Furthermore, the CuO@RF@Ag nanowires can also reveal a high-sensitivity SERS-active substrate for detecting amoxicillin (10−10 M) and 5-fluorouracil (10−7 M).
- CuO@RF@Ag nanowires exhibit a simple fabrication process, high SERS sensitivity, high reproducibility, high uniformity, and low detection limit, which are helpful for the practical application of SERS in different fields.
Reviewer 2 Report
My previous report indicated, between the minor points to be taken into account, some typos and errors that had to be corrected. To be more clear I made there some examples, but the needed text editing it is not limited to the sentences that I have expictly indicated. Authors have to well verify similar phrases alongside the entire manuscript. For example the sentence "Figure 3a reveals that the FETEM image of a CuO nanowire has been completely 110 decorated with the small sizes of Ag nanoparticles (2.5-12 nm)." must be corrected in "The FETEM image in Figure 3a reveals that the CuO nanowire has been completely decorated with small sizes Ag nanoparticles (2.5-12 nm)", or in the caption of figure 2 "were grown" must be substitued by "grown", and so on.
Author Response
My previous report indicated, between the minor points to be taken into account, some typos and errors that had to be corrected. To be more clear I made there some examples, but the needed text editing it is not limited to the sentences that I have expictly indicated. Authors have to well verify similar phrases alongside the entire manuscript. For example the sentence "Figure 3a reveals that the FETEM image of a CuO nanowire has been completely 110 decorated with the small sizes of Ag nanoparticles (2.5-12 nm)." must be corrected in "The FETEM image in Figure 3a reveals that the CuO nanowire has been completely decorated with small sizes Ag nanoparticles (2.5-12 nm)", or in the caption of figure 2 "were grown" must be substitued by "grown", and so on.
Response: Thanks for the pertinent and positive comments. We have amended this mistake in the revised manuscript.